

# Cost-effectiveness of antiviral therapy during late pregnancy to prevent perinatal transmission of hepatitis B virus

Wenjun Wang[1,2], Jingjing Wang[3], Shuangsuo Dang[1,2] and Guihua Zhuang[2]

[1] Department of Infectious Diseases, The Second Affiliated Hospital of Xi'an Jiaotong University, Xi'an, China
[2] Department of Epidemiology and Biostatistics, Medical School of Xi'an Jiaotong University, Xi'an, China
[3] Department of Pediatrics, The Second Affiliated Hospital of Xi'an Jiaotong University, Xi'an, China

Corresponding author
Shuangsuo Dang,
dangss212@yahoo.com

## ABSTRACT

**Background.** Hepatitis B virus (HBV) infections are perinatally transmitted from chronically infected mothers. Supplemental antiviral therapy during late pregnancy with lamivudine (LAM), telbivudine (LdT), or tenofovir (TDF) can substantially reduce perinatal HBV transmission compared to postnatal immunoprophylaxis (IP) alone. However, the cost-effectiveness of these measures is not clear.

**Aim.** This study evaluated the cost-effectiveness from a societal perspective of supplemental antiviral agents for preventing perinatal HBV transmission in mothers with high viral load ($>6 \log_{10}$ copies/mL).

**Methods.** A systematic review and network meta-analysis were performed for the risk of perinatal HBV transmission with antiviral therapies. A decision analysis was conducted to evaluate the clinical and economic outcomes in China of four competing strategies: postnatal IP alone (strategy IP), or in combination with perinatal LAM (strategy LAM + IP), LdT (strategy LdT + IP), or TDF (strategy TDF + IP). Antiviral treatments were administered from week 28 of gestation to 4 weeks after birth. Outcomes included treatment-related costs, number of infections, and quality-adjusted life years (QALYs). One- and two-way sensitivity analyses were performed to identify influential clinical and cost-related variables. Probabilistic sensitivity analyses were used to estimate the probabilities of being cost-effective for each strategy.

**Results.** LdT + IP and TDF + IP averted the most infections and HBV-related deaths, and gained the most QALYs. IP and TDF + IP were dominated as they resulted in less or equal QALYs with higher associated costs. LdT + IP had an incremental $2,891 per QALY gained (95% CI [$932–$20,372]) compared to LAM + IP (GDP per capita for China in 2013 was $6,800). One-way sensitivity analyses showed that the cost-effectiveness of LdT + IP was only sensitive to the relative risk of HBV transmission comparing LdT + IP with LAM + IP. Probabilistic sensitivity analyses demonstrated that LdT + IP was cost-effective in most cases across willingness-to-pay range of $6,800 ∼ $20,400 per QALY gained.

**Conclusions.** For pregnant HBV-infected women with high levels of viremia, supplemental use of LdT during late pregnancy combined with postnatal IP for infants is cost-effective in China.

## INTRODUCTION

Chronic hepatitis B virus (HBV) infection is a serious public health problem in China that affects ∼93 million people (7.18% of the population), representing a quarter of all HBV-infected people worldwide (*Liang et al., 2009*). HBV-related liver diseases and liver cancer are responsible for up to 300,000 deaths each year (*World Health Organization, 2013*). The economic burden of HBV infections in China alone is estimated to be in excess of 80 billion US dollars annually (*Chinese Center for Disease Control and Prevention, 0000*).

The predominant mode of HBV infection involves perinatal transmission from chronically infected mothers to their infants (*Gambarin-Gelwan, 2007*; *Patton & Tran, 2014*; *Stevens et al., 1975*). This can be suppressed by passive immunization with hepatitis B immunoglobulin (HBIG) administered within 12 h of birth and active immunization with a three-dose HBV vaccination series (*Lee et al., 2006*). The success of these immunoprophylaxis (IP) measures is demonstrated by the low prevalence of hepatitis B surface antigen (HBsAg) in children under five years of age in mainland China and the substantial reduction in HBV infection in Taiwan (*Liang et al., 2009*; *Ni et al., 2007*).

Despite these encouraging results, a proportion of infants still test positive for the HBsAg (*Lee et al., 2006*). The mother's viral load in serum is an important contributing factor, as infants born from HBeAg (hepatitis B e-antigen) positive mothers with high levels of viremia are at greatest risk for perinatal transmission (*Burgis et al., 2014*; *Burk et al., 1994*; *Del Canho et al., 1994*). HBV replication in pregnant mothers can safely be suppressed with nucleos(t)ide analogues, such as lamivudine (LAM), telbivudine (LdT), and tenofovir(TDF), (*Brown et al., 2012*; *Fontana, 2009*; *Wang et al., 2013*; *Yi, Liu & Cai, 2012*) which reduce perinatal transmission when administered during the third trimester (*Celen et al., 2013*; *Han et al., 2011*; *Lu et al., 2014*; *Pan et al., 2012*). Previous studies indicate that supplemental prenatal use of LAM is cost-effective compared with postnatal IP only (*Fan et al., 2014*; *Hung & Chen, 2011*; *Nayeri et al., 2012*; *Unal et al., 2011*). Although LdT and TDF are more potent, the costs for these treatments are higher. Therefore, the purpose of this study was to examine the cost-effectiveness of these treatments from a societal perspective for prevention of perinatal transmission of HBV in China.

## MATERIALS AND METHODS

### Model and strategies

Decision analysis software (TreeAge Pro 2012: TreeAge Software Inc., Williamstown, MA, USA) was used to estimate the clinical and economic outcomes for a hypothetical cohort of 10,000 pregnant Chinese women chronically infected with HBV. All women were considered positive for HBeAg with serum levels of HBV-DNA >6 $\log_{10}$ copies/mL. The following four strategies were compared: (1) postnatal IP for the infants only; (2) perinatal LAM and postnatal IP (LAM + IP); (3) perinatal LdT and postnatal IP (LdT + IP); and (4) perinatal TDF and postnatal IP (TDF + IP) (Fig. 1). Perinatal antiviral treatments were administered to the pregnant women beginning from week 28 of gestation to 4 weeks after birth. Perinatal transmission of HBV was defined as HBsAg positive of infants 6 months after birth.

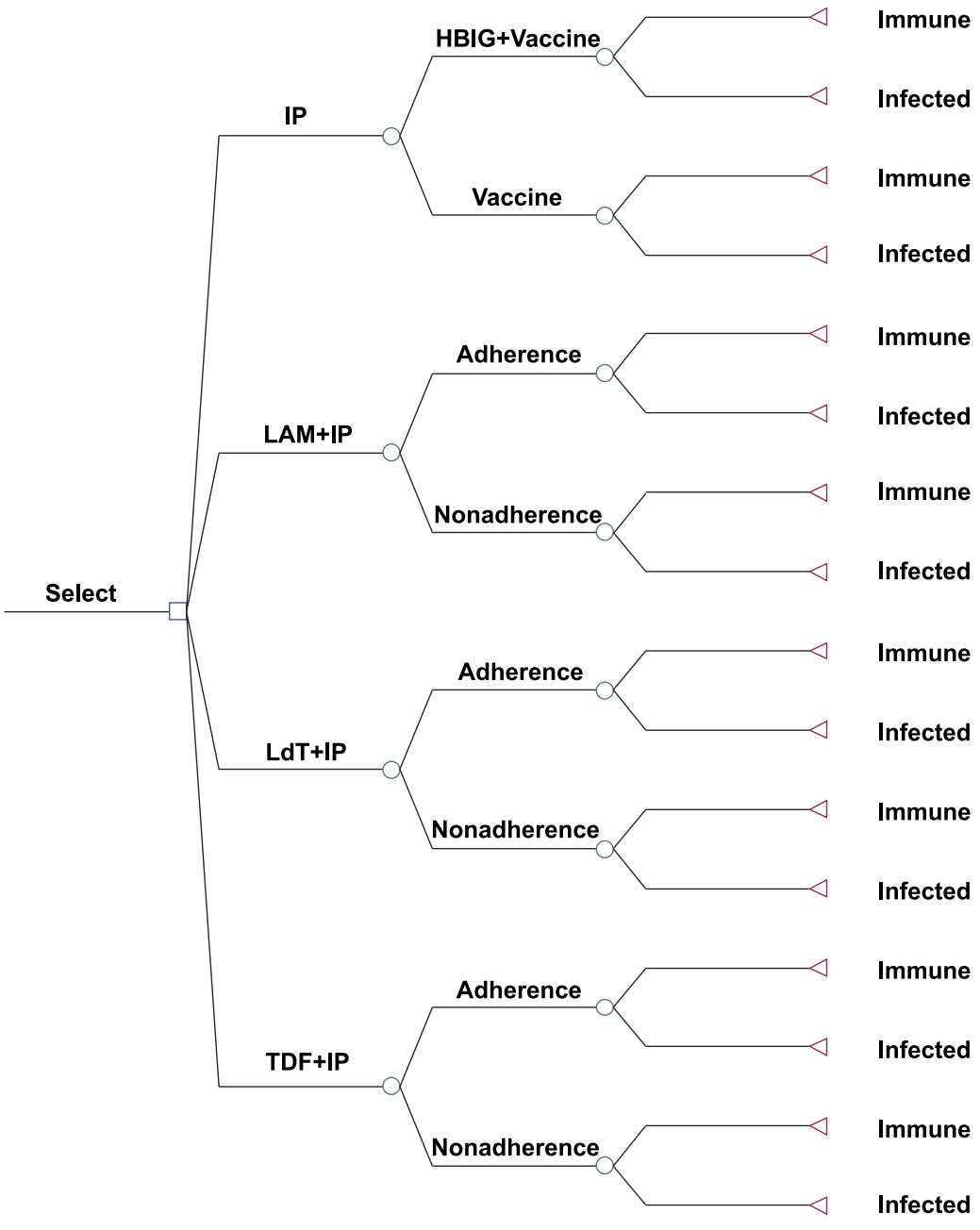

**Figure 1** **Decision tree model showing the four strategies for preventing perinatal hepatitis B transmission.** HBIG, hepatitis B immunoglobulin; IP, immunoprophylaxis; LAM, lamivudine; LdT, telbivudine; TDF, tenofovir.

If infants were infected by HBV, their lifetime clinical and economic outcomes were simulated using a Markov model based on a previous study (*Hutton, So & Brandeau, 2010*) (Fig. S1). This model was composed of the following eight states: normal alanine aminotransferase (ALT), elevated ALT, durable response, compensated cirrhosis, decompensated cirrhosis, hepatocellular carcinoma, liver transplantation, and death. The cycle length was one year and the cycle number was set to 73 corresponding to the life

expectancy of Chinese people (*National Health and Family Planning Commission of the People's Republic of China, 2013*).

## Systematic review and meta-analysis

A systematic review and network meta-analysis were performed to calculate odds ratios (ORs) for perinatal transmission of HBV; see Fig. S2 and Article S1. Briefly, PubMed, Embase, Cochrane Central Register of Controlled Trials, and the Chinese Biomedical Literature Database were searched to identify controlled studies addressing prenatal antiviral therapy. Although a number of studies evaluating LAM + IP or LdT + IP versus IP were identified, very few compared LAM + IP and LdT + IP directly (Fig. S3 and Table S1) (*Feng & Zhang, 2007*; *Guo et al., 2008*; *Xu et al., 2009*; *Zhang, 2010*; *Zhao et al., 2010*; *Guo & Zhang, 2011*; *Han et al., 2011*; *Yao et al., 2011*; *Zhou et al., 2011*; *Pan et al., 2012*; *Peng et al., 2012*; *Wang, 2012*; *Wang & Zhao, 2012*; *Bai, Shang & Li, 2013*; *Jiang et al., 2013*; *Sun et al., 2013*; *Wu et al., 2013*; *Zeng et al., 2013*; *Yu et al., 2014*; *Zhang et al., 2014*). As a result, a network meta-analysis was deemed more suitable (*Caldwell, Ades & Higgins, 2005*). All clinical estimates and their ranges are shown in Table 1.

The reported ORs of HBV transmission were 0.21 (95% CI [0.12–0.38]) for LAM + IP versus IP, and 0.55 (95% CI [0.24–1.29]) for LdT + IP versus LAM + IP. The corresponding relative risks (RRs) of HBV transmission were then calculated using the formula: $RR = OR/((1 - P_0) + (P_0 \times OR))$, where $P_0$ indicates the incidence of HBV transmission in the IP (for LAM + IP versus IP) or LAM + IP (for LdT + IP versus LAM + IP) groups based on the weighted mean by the study sample size; $P_0 = 0.117$ and 0.027, respectively. Furthermore, the 95% CIs of the ORs were transformed to RR form for sensitivity analyses. An equal risk of HBV transmission was assumed for TDF + IP and LdT + IP in the base-case analyses based on the antiviral potencies of TDF and LdT and two small-sized studies (*Celen et al., 2013*; *Pan et al., 2012*; *Lai et al., 2007*; *Marcellin et al., 2008*). For sensitivity analyses, it was assumed that the risk for TDF + IP was not likely to be less than half of that for LdT + IP. The range of transmission rates for the IP group reported in each study were recorded and included in sensitivity analyses.

Although IP is the current recommendation in China to interrupt perinatal hepatitis B transmission, (*Chinese Society of Hepatology and Chinese Society of Infectious Diseases, 2011*; *Chinese Society of Gynaecology and Obstetrics, 2013*) the compliance rate varies across regions (*Cui et al., 2013*; *Guo et al., 2010*; *Hu et al., 2012*). Using estimates of the proportions of children receiving HBV vaccine plus HBIG in three regions reported by *Cui et al. (2013)* the overall proportion was calculated by weighting the sample size from each region.

The high reported adherence to antiviral treatment in pregnant mothers (*Celen et al., 2013*; *Feng & Zhang, 2007*; *Guo et al., 2008*; *Xu et al., 2009*; *Zhang, 2010*; *Zhao et al., 2010*; *Guo & Zhang, 2011*; *Han et al., 2011*; *Yao et al., 2011*; *Zhou et al., 2011*; *Pan et al., 2012*; *Peng et al., 2012*; *Wang, 2012*; *Wang & Zhao, 2012*; *Bai, Shang & Li, 2013*; *Jiang et al., 2013*; *Sun et al., 2013*; *Wu et al., 2013*; *Zeng et al., 2013*; *Yu et al., 2014*; *Zhang et al., 2014*) is likely due to a variety of factors, including a strong desire to protect their infants against HBV, the safety of the antiviral agents, short duration of treatment, and low incidence of

**Table 1  Clinical variables for base-case and sensitivity analysis.**

| Variable | Base case | Range | Ref. |
|---|---|---|---|
| Probability of perinatal hepatitis B transmission with IP | 11.7% | 2.8–42.3% | *Celen et al., 2013; Feng & Zhang, 2007; Guo et al., 2008; Xu et al., 2009; Zhang, 2010; Zhao et al., 2010; Guo & Zhang, 2011; Han et al., 2011; Yao et al., 2011; Zhou et al., 2011; Pan et al., 2012; Peng et al., 2012; Wang, 2012; Wang & Zhao, 2012; Bai, Shang & Li, 2013; Jiang et al., 2013; Sun et al., 2013; Wu et al., 2013; Zeng et al., 2013; Yu et al., 2014; Zhang et al., 2014* |
| Relative risk of perinatal hepatitis B transmission | | | |
| Vaccination *vs*. IP | 1.85 | 1.37–2.44 | *Lee et al., 2006* |
| LAM + IP *vs*. IP | 0.23 | 0.13–0.41 | *Feng & Zhang, 2007; Guo et al., 2008; Xu et al., 2009; Zhang, 2010; Zhao et al., 2010; Guo & Zhang, 2011; Han et al., 2011; Yao et al., 2011; Zhou et al., 2011; Pan et al., 2012; Peng et al., 2012; Wang, 2012; Wang & Zhao, 2012; Bai, Shang & Li, 2013; Jiang et al., 2013; Sun et al., 2013; Wu et al., 2013; Zeng et al., 2013; Yu et al., 2014; Zhang et al., 2014* |
| LdT + IP *vs*. LAM + IP | 0.56 | 0.25–1[a] | *Feng & Zhang, 2007; Guo et al., 2008; Xu et al., 2009; Zhang, 2010; Zhao et al., 2010; Guo & Zhang, 2011; Han et al., 2011; Yao et al., 2011; Zhou et al., 2011; Pan et al., 2012; Peng et al., 2012; Wang, 2012; Wang & Zhao, 2012; Bai, Shang & Li, 2013; Jiang et al., 2013; Sun et al., 2013; Wu et al., 2013; Zeng et al., 2013; Yu et al., 2014; Zhang et al., 2014* |
| TDF + IP *vs*. LdT + IP | 1 | 0.5–1 | *Celen et al., 2013; Pan et al., 2012; Lai et al., 2007; Marcellin et al., 2008* |
| Compliance rate of IP | 44.0% | 37.6–86.0% | *Cui et al., 2013; Guo et al., 2010; Hu et al., 2012* |
| Adherence to antiviral treatment during pregnancy | 98.4% | 80.0–100% | *Feng & Zhang, 2007; Guo et al., 2008; Xu et al., 2009; Zhang, 2010; Zhao et al., 2010; Guo & Zhang, 2011; Han et al., 2011; Yao et al., 2011; Zhou et al., 2011; Pan et al., 2012; Peng et al., 2012; Wang, 2012; Wang & Zhao, 2012; Bai, Shang & Li, 2013; Jiang et al., 2013; Sun et al., 2013; Wu et al., 2013; Zeng et al., 2013; Yu et al., 2014; Zhang et al., 2014* |
| Natural history parameters | | | |
| Normal ALT to elevated ALT | 0.2% | 0.1–0.2% | *Hutton, So & Brandeau, 2010* |
| Normal ALT to HCC | 0.3% | 0.2–0.5% | *Hutton, So & Brandeau, 2010* |
| Chronic HBV with elevated ALT to compensated cirrhosis | 3.8% | 0.5–12.3% | *Hutton, So & Brandeau, 2010* |
| Chronic HBV with elevated ALT to HCC | 1.5% | 0.5–9.5% | *Hutton, So & Brandeau, 2010* |
| Durable virologic response while on treatment | 15.0% | 5.0–30.0% | *Hutton, So & Brandeau, 2010* |
| Receiving treatment with durable response | 50.0% | 0.0–100% | *Hutton, So & Brandeau, 2010* |
| Durable response relapse to elevated ALT | 7.0% | 2.0–15.0% | *Hutton, So & Brandeau, 2010* |
| Durable response relapse to HCC | 0.3% | 0.2–0.5% | *Hutton, So & Brandeau, 2010* |
| Compensated to decompensated cirrhosis | 7.0% | 3.0–10.0% | *Hutton, So & Brandeau, 2010* |
| Mortality from compensated cirrhosis | 4.8% | 2.0–13.1% | *Hutton, So & Brandeau, 2010* |
| Mortality from decompensated cirrhosis | 17.3% | 5.8–22.1% | *Hutton, So & Brandeau, 2010* |
| Cirrhosis to HCC | 3.3% | 1.0–11.3% | *Hutton, So & Brandeau, 2010* |
| Liver transplantation for decompensated cirrhosis | 1.5% | 0.0–40.0% | *Hutton, So & Brandeau, 2010* |
| Mortality from HCC | 40.0% | 32.0–47.3% | *Hutton, So & Brandeau, 2010* |

**Table 1** (*continued*)

| Variable | Base case | Range | Ref. |
|---|---|---|---|
| Liver transplantation for HCC | 0.1% | 0.0–40.0% | *Hutton, So & Brandeau, 2010* |
| Mortality first year after liver transplantation | 15.0% | 7.5–30.0% | *Hutton, So & Brandeau, 2010* |
| Mortality second and subsequent years after liver transplantation | 1.5% | 0.8–3.0% | *Hutton, So & Brandeau, 2010* |
| Health-state utility weights for quality of life adjustments | | | |
| Normal ALT | 1.00 | 0.95–1.00 | *Hutton, So & Brandeau, 2010* |
| Elevated ALT | 0.99 | 0.90–1.00 | *Hutton, So & Brandeau, 2010* |
| Durable response | 1.00 | 0.90–1.00 | *Hutton, So & Brandeau, 2010* |
| Compensated cirrhosis | 0.80 | 0.70–0.93 | *Hutton, So & Brandeau, 2010* |
| Decompensated cirrhosis | 0.60 | 0.50–0.70 | *Hutton, So & Brandeau, 2010* |
| HCC | 0.73 | 0.50–0.80 | *Hutton, So & Brandeau, 2010* |
| Liver transplantation | 0.86 | 0.70–0.90 | *Hutton, So & Brandeau, 2010* |

**Notes.**

ALT, alanine aminotransferase; HBV, hepatitis B virus; HCC, hepatocellular carcinoma; IP, immunoprophylaxis; LAM, lamivudine; LdT, telbivudine; TDF, tenofovir.

[a]Prevention efficacy of LAM was assumed to not be superior to LdT.

drug resistance (*Brown et al., 2012*; *Fontana, 2009*; *Wang et al., 2013*; *Yi, Liu & Cai, 2012*; *Ayres et al., 2014*; *Ho & Tran, 2011*). For those who did not fully comply, the risk of HBV transmission was assumed to equal that of IP alone. For mothers who chose antiviral strategies, 100% coverage of IP for their infants was assumed, regardless of their adherence to antiviral treatment.

Under the National Immunization Program, free HBV vaccinations are available to all Chinese infants, covering 97.6% of infants born from HBsAg-positive mothers in 2009 (*Cui et al., 2013*). Therefore, it was assumed that infants who did not receive HBIG at birth were covered by the HBV vaccination. According to a meta-analysis, the risk of HBV transmission with vaccination only is nearly twice that with vaccination plus HBIG (RR 1.85, 95% CI [1.37–2.44]) (*Lee et al., 2006*).

We obtained from a published study the natural history estimates of chronic HBV infection and health-state utility weights for calculating quality-adjusted life years (QALYs) (*Hutton, So & Brandeau, 2010*). An age-specific death rate table was used to estimate the death probabilities for HBV-infected people without complications and uninfected people (*National Bureau of Statistics of China, 2013a*).

## Costs

For cost analyses, the costs of HBV-marker tests (inclusive of HBeAg and HBsAg tests), HBV-DNA quantification, antiviral drugs, vaccination, and HBIG were included. The actual cost of HBV-DNA quantification for one mother who is going to receive antiviral treatment was calculated as $n+1$ times the cost of one test, where $n$ indicates the ratio of the proportion of mothers with HBV-DNA < 6 $\log_{10}$ copies/mL to those with HBV-DNA > 6 $\log_{10}$ copies/mL, as previously described (*Fung et al., 2011*). The cost of only one dose of HBIG was included, as a meta-analysis showed that multiple doses of HBIG are not superior to a single dose of HBIG in preventing perinatal transmission of HBV (*Lee et al., 2006*).

The costs of vaccination and HBIG were obtained from published studies, (*Guo et al., 2012*; *Shi & Zhang, 2013*), LAM and LdT from Price Bureau of Shaanxi Province, (*Price*

**Table 2   Cost variables for base-case and sensitivity analysis.**

| Variable | Base case (USD) | Range (USD) | Ref. |
|---|---|---|---|
| Hepatitis B vaccination, three times | $3.0 | $1.5–6.0 | *Shi & Zhang, 2013* |
| Hepatitis B immunoglobulin administration | $40.0 | $20.0–80.0 | *Guo et al., 2012*; *Shi & Zhang, 2013* |
| LAM, daily[a] | $2.5 | $1.3–5.0 | *Price Bureau of Shaanxi Province, 0000* |
| LdT, daily[a] | $3.6 | $1.8–7.2 | *Price Bureau of Shaanxi Province, 0000* |
| TDF, daily[a] | $8.7 | $4.4–17.4 | *Kangdele Pharmacy, 2014* |
| Ratio of proportion of mothers with <6 to>6 $\log_{10}$ copies/mL HBV-DNA | 0.136 | 0.068–0.273 | *Fung et al., 2011* |
| HBV-DNA quantification | $16.3 | $8.2–32.6 | *Price Bureau of Shaanxi Province, 0000* |
| HBV-marker test | $3.3 | $1.6–6.6 | *Price Bureau of Shaanxi Province, 0000* |
| Chronic hepatitis B, annual[b] | $1,780 | $890–3,560 | *Hu & Chen, 2009* |
| Compensated cirrhosis, annual | $2,759 | $1,380–5,518 | *Hu & Chen, 2009* |
| Decompensated cirrhosis, annual | $5,130 | $2,565–10,260 | *Hu & Chen, 2009* |
| Hepatocellular carcinoma, annual | $7,302 | $3,651–14,604 | *Hu & Chen, 2009* |
| Liver transplantation, first year | $37,458 | $18,729–74,916 | *Chen, Yao & Chen, 2007* |
| Liver transplantation, second and subsequent years, annual | $3,276 | $1,638–6,552 | *Chen, Yao & Chen, 2007* |
| Discount rate, annual | 3% | 0–5% | – |

Notes.

HBV, hepatitis B virus; LAM, lamivudine; LdT, telbivudine; TDF, tenofovir; USD, US Dollar.

[a]Administered from week 28 of gestation to 4 weeks after delivery.

[b]50% of patients with durable response were assumed to continue receiving treatment.

*Bureau of Shaanxi Province, 0000*), and TDF from *Kangdele Pharmacy (2014)*. The costs for HBV-related diseases incorporated direct medical costs and non-medical costs for physician visits, medications, lab tests, and transportation, and indirect costs for work loss, as reported in published studies (*Chen, Yao & Chen, 2007*; *Hu & Chen, 2009*).

All cost estimates were converted to US Dollars according to the 2013 conversion rate (1 US Dollar = 6.13 Chinese Yuan) using the medical care component of the Consumer Price Index and discounting costs and QALYs to 2013 amounts at a rate of 3% per year (*National Bureau of Statistics of China, 2013b*). All cost estimates and their ranges are listed in Table 2.

## Outcomes

Short-term outcomes evaluated clinical and economic outcomes during the period from the initiation of the strategies to 6 months after birth, including the number of infections, incremental infections averted, prophylaxis costs, incremental costs, and the corresponding incremental cost-effectiveness ratio (ICER). Long-term outcomes evaluated lifetime clinical and economic outcomes under four strategies, including HBV-related death, QALYs, incremental QALYs, lifetime costs, incremental lifetime costs, and the corresponding ICER. The ICER was used to compare alternative strategies after eliminating those that were dominated (more costly and less effective). It was calculated as the incremental cost divided by the incremental health benefit (e.g., infections averted and QALYs gained) for one strategy compared to the next less-costly strategy. The cost-effectiveness analyses were conducted from a societal perspective in accordance with the World Health Organization

**Table 3  Short-term outcomes.**

| Strategy | Infections (*n*) | Incremental infections averted (*n*) | Cost (USD) | Incremental cost (USD) | ICER |
|---|---|---|---|---|---|
| IP | 1,727 | – | $239,000 | – | – |
| LAM + IP | 284 | 1,443 | $3,078,568 | $2,839,568 | 1,967 |
| LdT + IP | 167 | 117 | $4,147,944 | $1,069,376 | 9,178 |
| TDF + IP | 167 | 0 | $9,105,960 | $4,958,016 | – |

**Notes.**
ICER, incremental cost-effectiveness ratio (US dollars per incremental infection averted); IP, immunoprophylaxis; LAM, lamivudine; LdT, telbivudine; TDF, tenofovir; USD, US Dollar.

recommendations, and cost-effectiveness thresholds were based on the gross domestic product (GDP) per capita: highly cost-effective (ICER < GDP per capita); cost-effective (GDP per capita < ICER < 3 × GDP per capita); and not cost-effective (ICER > 3 × GDP per capita) (*World Health Organization, 2014*). The GDP per capita for China in 2013 was approximately $6,800 (*National Bureau of Statistics of China, 2013b*).

## Sensitivity analyses

One- and two-way sensitivity analyses were performed to identify influential clinical and cost-related variables. In addition, a probabilistic sensitivity analysis was conducted based on a second-order Monte Carlo simulation (*Geisler, 2011*). All variables from Table 1 that were put into the simulation were assumed to follow a triangle distribution (likeliest, minimum, and maximum values) (*Kanwal et al., 2005*). A total of 1000 trials were simulated, and cost-effectiveness acceptability curves for competing strategies were constructed.

# RESULTS

## Base-case analyses

Antiviral strategies prevented more perinatal hepatitis B transmissions than IP alone. The fewest number of infections occurred with LdT + IP and TDF + IP (Fig. 2). Short-term cost was increased with effectiveness, except for TDF + IP, which was equally as effective as LdT + IP, but at more than twice the cost (Table 3).

In the long-term, the more effective strategies resulted in fewer instances of hepatocellular carcinoma and HBV-related death, and increased the QALYs (Table 4). IP and TDF + IP were dominated because they resulted in the same or fewer QALYs but at a comparatively higher cost. LdT + IP had an incremental $2,891 per QALY gained (95% CI [−$932 ∼ $20,372]) compared to LAM + IP.

## Sensitivity analyses

One-way sensitivity analyses were performed across the ranges of all clinical and cost-related variables. The cost-effectiveness of LdT + IP was only sensitive to one of them, the RR of HBV transmission in comparison to LAM + IP. If the RR is above 0.92, LdT + IP would be not cost-effective.

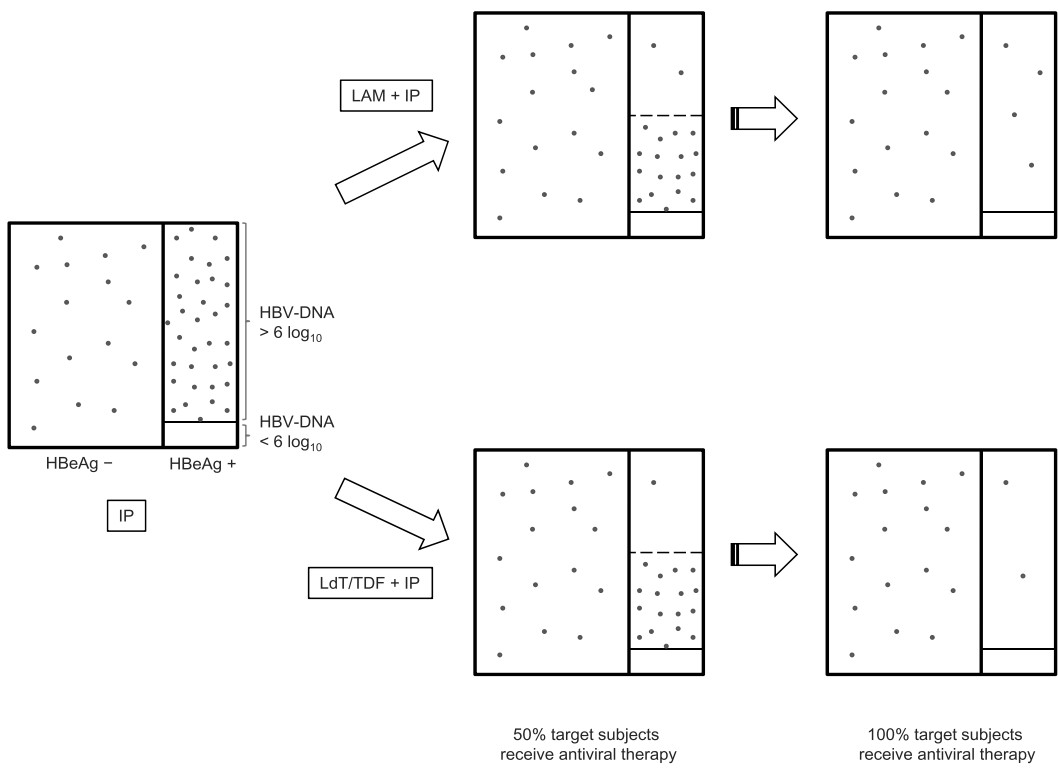

**Figure 2** **Effect of antiviral strategies on reducing perinatal hepatitis B transmission.** The area of each rectangle represents the proportion of pregnant women, and the density of dots represents the probability of perinatal transmission of HBV. The unit of HBV-DNA is copies/mL. HBeAg, hepatitis B e antigen; HBV, hepatitis B virus; IP, immunoprophylaxis; LAM, lamivudine; LdT, telbivudine; TDF, tenofovir.

**Table 4** **Long-term outcomes.**

| Strategy | HCC (n) | HBV-related deaths (n) | QALYs | Incremental QALYs | Cost (USD) | Incremental cost (USD) | ICER (95% CI) |
|---|---|---|---|---|---|---|---|
| IP | 385 | 304 | 292,167 | – | $4,897,077 | – | Dominated |
| TDF + IP | 37 | 29 | 295,664 | – | $9,556,428 | – | Dominated |
| LAM + IP | 63 | 50 | 295,403 | – | $3,843,301 | – | – |
| LdT + IP | 37 | 29 | 295,664 | 261 | $4,598,412 | $755,111 | $2,891 (−$932~$20,372) |

**Notes.**

CI, confidence interval; HBV, hepatitis B virus; HCC, hepatocellular carcinoma; ICER, incremental cost-effectiveness ratio (US Dollar per quality-adjusted life year gained); IP, immunoprophylaxis; LAM, lamivudine; LdT, telbivudine; QALY, quality-adjusted life year; TDF, tenofovir; USD, US Dollar.

A two-way sensitivity analysis showed that LdT + IP remains cost-effective even when its cost is doubled and that of LAM halved. When TDF + IP protects more infants than LdT + IP and simultaneously the cost of TDF goes down, TDF + IP may become cost-effective (Fig. S4). With lower transmission risk of TDF + IP compared with LdT + IP, TDF + IP may also become cost-effective with changes in additional variables, such as the probabilities of HBV transmission for competing strategies, the discount rate, the cost of LdT, and the utility weight for the state of chronic HBV infection with normal ALT levels.

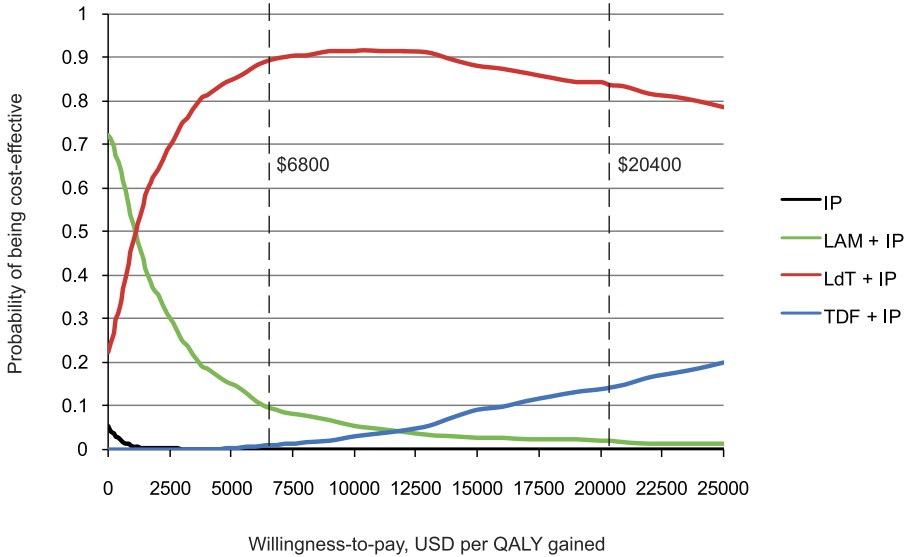

**Figure 3 Acceptability curves of the four strategies for preventing perinatal hepatitis B transmission.** Chinese gross domestic product per capita was approximately $6,800 in 2013. IP, immunoprophylaxis; LAM, lamivudine; LdT, telbivudine; TDF, tenofovir; USD, US Dollar; QALY, quality-adjusted life year.

Acceptability curves constructed from probabilistic sensitivity analyses showed that LdT + IP was highly cost-effective in 89.8% of the trials under a willingness-to-pay threshold of $6,800, and cost-effective in 83.7% under a threshold of $20,400 (Fig. 3). As the price of TDF is likely to change in the near future, acceptability curves were simulated for a series of TDF cost from $8 to $3.6 daily. As the cost of TDF decreases, TDF + IP would become the most cost-effective strategy (Fig. S5).

## DISCUSSION

Antiviral treatment as prophylaxis for perinatal hepatitis B transmission is recommended in updated guidelines from the European Association for the Study of the Liver, Asian Pacific Association for the Study of the Liver, and the National Institute for Health and Care Excellence (*European Association For The Study Of The Liver, 2012*; *Liaw et al., 2012*; *Sarri et al., 2013*). However, there is no consensus regarding which antiviral agent is most appropriate. Only one agent, LAM, was included in previous cost-effectiveness analyses of high-income areas only (*Fan et al., 2014*; *Hung & Chen, 2011*; *Nayeri et al., 2012*; *Unal et al., 2011*). The results of the present analyses confirm that conventional IP alone is not cost-effective under any circumstance, and further demonstrate that supplemental use of LdT is highly cost-effective, and preferable over LAM or TDF in China.

Implementation of the National Immunization Program that provides HBV vaccinations to all neonates and extends HBIG administration to those whose mothers are infected with HBV has greatly reduced the prevalence of HBV (*Liang et al., 2009*; *Cui et al., 2013*). Thus, antiviral treatment during late pregnancy in women with a high viral load will have a further impact, not only in terms of health benefits but also in terms of economic costs. Supplementation of IP with LdT or LAM will dramatically reduce the overall number
of HBV infections, and reduce perinatal transmission by at least half. Thus, one could expect the prevalence of HBV to be <0.5% among those at the highest risk for infection, i.e., children under five years of age. The initial additional cost of antiviral treatment will be outweighed by the reduced economic burden of HBV-related diseases.

From a cost-effectiveness perspective, China will benefit from the use of LAM or LdT antiviral strategies to prevent HBV transmission. Indeed, Chinese hepatologists in tertiary hospitals have been prescribing these agents to pregnant women with chronic HBV infection for several years (*Li et al., 2003*; *Zhang & Wang, 2009*). However, pregnant infected women are more likely to visit the obstetrics clinic first, and may not seek consultation from hepatologists. The recent guideline from the Chinese Society of Obstetrics and Gynecology recommends against antiviral treatment as prophylaxis of perinatal transmission of HBV (*Chinese Society of Gynaecology and Obstetrics, 2013*). A survey of Chinese obstetricians showed that only 11.7% agree with antiviral treatment during pregnancy (*Hu et al., 2013*). Delayed update of evidence and differences in research interest between obstetricians and hepatologists may contribute to the above knowledge and practices among gynaecology staff. The findings presented in this study highlight the critical need to further summarize the clinical evidence and evaluate the cost-effectiveness of antiviral treatment to prevent perinatal transmission of HBV.

Sensitivity analyses indicate that the tradeoff between the use of LdT and LAM is influenced by the RR of HBV transmission. However, LdT remains more cost-effective than LAM when transmission risk is reduced by >8%. The sensitivity analyses also indicate that more information is needed concerning the efficacy and cost of TDF. It is difficult to determine an added benefit of TDF when studies with large sample sizes show a 100% efficacy with LdT (*Han et al., 2011*; *Yu et al., 2014*; *Zhang et al., 2014*). As a result, the cost of TDF becomes the main variable affecting its cost-effectiveness. TDF is less expensive than LdT in Western countries such as Spain, (*Buti et al., 2009*), where TDF would be considered more cost-effective assuming an equal efficacy to LdT. On this point, our results are consistent with the recent guidelines of the National Institute for Health and Care Excellence (*Sarri et al., 2013*). TDF will also become cost-effective in China as its cost decreases.

There are several important strengths of the current study. This study is the first decision analysis that compares all currently available antiviral agents for the prevention of perinatal transmission of HBV. In the model used for the analyses, the only variables that differed among each antiviral strategy were the prevention efficacy and the cost. Consistent with previous studies, (*Fan et al., 2014*; *Hung & Chen, 2011*; *Nayeri et al., 2012*; *Unal et al., 2011*) LAM was found to be cost-effective compared with the conventional IP, indicating that the findings concerning LdT and TDF are similarly valid and reliable. In addition, a systematic review was performed to summarize the prevention efficacy of each antiviral agent. Included in the current analyses are two newly published high-quality studies with large sample sizes (*Yu et al., 2014*; *Zhang et al., 2014*). Finally, a network meta-analysis was used to evaluate the efficacies of each strategy, which combines direct and indirect comparisons and is superior to traditional pairwise analyses when there is an insufficient number of direct comparison studies (*Caldwell, Ades & Higgins, 2005*).

There are some limitations of this study that should be mentioned. First, the risk of side effects when using antiviral agents during pregnancy was not considered. However, controlled studies demonstrate that LAM, LdT, or TDF do not increase the number of birth defects or complications (*Brown et al., 2012*; *Celen et al., 2013*; *Feng & Zhang, 2007*; *Guo et al., 2008*; *Xu et al., 2009*; *Zhang, 2010*; *Zhao et al., 2010*; *Guo & Zhang, 2011*; *Han et al., 2011*; *Yao et al., 2011*; *Zhou et al., 2011*; *Pan et al., 2012*; *Peng et al., 2012*; *Wang, 2012*; *Wang & Zhao, 2012*; *Bai, Shang & Li, 2013*; *Jiang et al., 2013*; *Sun et al., 2013*; *Wu et al., 2013*; *Zeng et al., 2013*; *Yu et al., 2014*; *Zhang et al., 2014*). Second, there is no consensus on when antiviral treatment should be withdrawn, though most studies chose four weeks after delivery as the end time. Except for a small fraction of mothers continuing to receive the treatment because of elevated ALT levels, most mothers discontinued the treatment as planned with no associated severe adverse events, though the results for mothers were not included in the current model. Third, there is no consensus on the threshold of HBV-DNA level for initiating antiviral treatment in pregnant women. Like most studies of relevant topic, the analysis used the threshold of 6 $\log_{10}$ copies/mL. Lastly, the analysis focused only on pregnant women positive for HBeAg. Further studies evaluating strategies to prevent perinatal transmission of HBV from HBeAg-negative mothers are needed (*Kubo et al., 2014*; *Pan et al., 2012*).

In conclusion, this cost-effectiveness analysis focusing on pregnant women positive for HBeAg with HBV-DNA > 6 $\log_{10}$ copies/mL suggests that supplemental use of LdT during late pregnancy combined with IP for their infants is cost-effective in China.

### Abbreviations

| | |
|---|---|
| ALT | alanine aminotransferase |
| CI | confidence interval |
| HBeAg | hepatitis B e antigen |
| HBIG | hepatitis B immunoglobulin |
| HBsAg | hepatitis B surface antigen |
| HBV | hepatitis B virus |
| ICER | incremental cost-effectiveness ratio |
| IP | immunoprophylaxis |
| LAM | lamivudine |
| LdT | telbivudine |
| OR | odds ratio |
| QALY | quality-adjusted life year |
| RR | relative risk |
| TDF | tenofovir |

### Funding

The authors received no funding for this work.

## Competing Interests

The authors declare there are no competing interests.

## Author Contributions

- Wenjun Wang conceived and designed the experiments, performed the experiments, analyzed the data, contributed reagents/materials/analysis tools, wrote the paper, prepared figures and/or tables, reviewed drafts of the paper.
- Jingjing Wang and Guihua Zhuang performed the experiments, analyzed the data, contributed reagents/materials/analysis tools, reviewed drafts of the paper.
- Shuangsuo Dang conceived and designed the experiments, wrote the paper, reviewed drafts of the paper.

## Data Availability

The research in this article did not generate any raw data.

## Supplemental Information

Supplemental information for this article can be found online at http://dx.doi.org/10.7717/peerj.1709#supplemental-information.

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
