# Peer review of "Cost-effectiveness of antiviral therapy during late pregnancy to prevent perinatal transmission of hepatitis B virus"

_PeerJ, doi:10.7717/peerj.1709_

## Round 0.1 · original submission · Major Revisions

As we got a lot of difficulties in getting all the reviews we needed, I reviewed the paper my-self and add these comments to the comments of the reviewer.
* * *
PJ-2015:06:5399:0:1
Cost-effectiveness of antiviral therapy during late pregnancy to prevent perinatal transmission of hepatitis B virus
Received: July 11 2015
Reviewed by Editor: Aug 19 2015

General statement

We apologize for the long delay in the handling of this paper. We contacted many reviewers but had only one who eventually accepted the task. I therefore undertook to review the paper my-self to second the reviewer who accepted and filed a report.

Globally, I agree with the reviewer’s report but would like to add the followings

1. The basis for this type of work and for its conclusions is strictly linked to the commercial prize of the treatments. While this is clearly acknowledged by the authors in their discussion, it is completely ignored in the main parts of the paper. This, I wonder whether the data of Figure 3 (and the conclusions of the paper) would not be completely different if the acquisition prices of the various options would be modified. While fixed prices are important for a political or economical decision, this is not true in Science. Thus, I’d very much like to see simulations presented for varying prices of the options selected by the authors. This is important, as prices are likely to change in the near future, and such simulations could help in pushing Industry to the appropriate direction. It will also give to the study a more normative aspect (i.e. showing how the model can be applied to different conditions). I consider this a must in the context of a scientific paper.

2. The paper has TOO many acronyms (and some of them like Ltd are not very intuitive). Please, try to reduce them and/or provide a list of acronyms that can be placed before the Introduction so that the reader can easily refer to it to understand your jargon. But, please, remember that acronyms should be used to facilitate the reading of the paper, NOT the work of the author.
* * *
·

Basic reporting

This article is nicely structured, the content is well underbuilt and the article studies a relevant theme.
However, the abstract is not well structured and should be rewritten to reflect the real content of the manuscript.

Experimental design

The required standards are met.

Validity of the findings

Data presentation is good.
Optional: in order to make the results presentation more robust, an incertainty parameter around the ICER could also be reported

---

## Round 0.2 · accepted · Accept

I apologize for the delay but I wished to re-review the paper carefully myself.

Thank you for your corrections. The paper looks quite nice now.

One point, however: please check carefully for minor mistakes. For instance, I guess that on page 6 "(3) perinatal LdT and postnatal IP (LAM+IP)" should be "(3) perinatal LdT and postnatal IP (LdT+IP)".